# Temporin-SHa and Its Analogs as Potential Candidates for the Treatment of *Helicobacter pylori*

**DOI:** 10.3390/biom9100598

**Published:** 2019-10-11

**Authors:** Hamza Olleik, Elias Baydoun, Josette Perrier, Akram Hijazi, Josette Raymond, Marine Manzoni, Lucas Dupuis, Ghislain Pauleau, Yvain Goudard, Bruno de La Villéon, Géraldine Goin, Philippe Sockeel, Muhammad Iqbal Choudhary, Eric Di Pasquale, Muhammad Nadeem-ul-Haque, Hunain Ali, Arif Iftikhar Khan, Farzana Shaheen, Marc Maresca

**Affiliations:** 1Aix Marseille Univ, CNRS, Centrale Marseille, iSm2, 13397 Marseille, France; 2Department of Biology, American University of Beirut, Beirut-1107 2020, Lebanon; 3Doctoral School of Science and Technology, Research Platform for Environmental Science (PRASE), Lebanese University, Beirut 5, Lebanon; 4Université Paris 5, Hôpital Cochin, Service de bactériologie, 75014 Paris, France; 5Departement of Digestive, Endocrine and Metabolic Surgery, Hôpital Laveran, Military Health Service, 13013 Marseille, France; 6H.E.J. Research Institute of Chemistry, International Center for Chemical and Biological Sciences, University of Karachi, Karachi-75270, Pakistan; 7Aix-Marseille Univ, CNRS, INP, Inst Neurophysiopathol, 13005 Marseille, France

**Keywords:** *Helicobacter pylori*, antimicrobial peptides, temporin-SHa, human gastric explant, clinical strain, bacterial resistance

## Abstract

*Helicobacter**pylori* is one of the most prevalent pathogens colonizing 50% of the world’s population and causing gastritis and gastric cancer. Even with triple and quadruple antibiotic therapies, *H. pylori* shows increased prevalence of resistance to conventional antibiotics and treatment failure. Due to their pore-forming activity, antimicrobial peptides (AMP) are considered as a good alternative to conventional antibiotics, particularly in the case of resistant bacteria. In this study, temporin-SHa (a frog AMP) and its analogs obtained by Gly to Ala substitutions were tested against *H. pylori*. Results showed differences in the antibacterial activity and toxicity of the peptides in relation to the number and position of D-Ala substitution. Temporin-SHa and its analog NST1 were identified as the best molecules, both peptides being active on clinical resistant strains, killing 90–100% of bacteria in less than 1 h and showing low to no toxicity against human gastric cells and tissue. Importantly, the presence of gastric mucins did not prevent the antibacterial effect of temporin-SHa and NST1, NST1 being in addition resistant to pepsin. Taken together, our results demonstrated that temporin-SHa and its analog NST1 could be considered as potential candidates to treat *H. pylori*, particularly in the case of resistant strains.

## 1. Introduction

*Helicobacter pylori* is a Gram-negative microaerophilic bacterium able to establish a lifelong infection in humans after its acquisition during childhood [1]. *H. pylori* is one of the most prevalent pathogens, which colonizes 50% of the world’s population (and more than 90% of the population in a developing country) and survives in the human stomach if not treated. Gastric chronic infection with *H. pylori* causes many gastrointestinal diseases including peptic ulcers and gastritis in 10% of patients. Importantly, *H. pylori* infection has been associated to gastric carcinoma in 1% of the case, causing around 700,000 deaths annually worldwide [2]. In addition to health issues, *H. pylori* infection has important economic consequences with approximately six billion losses annually [2]. Prevention of health and economic complications associated with *H. pylori* depends on its successful eradication [3,4].

Although vaccine is a way to prevent initial *H. pylori* infection, antibiotics are actually the best and mostly used strategy to cure installed infection. The first-line treatment developed in 1997 was based on a seven day triple therapy with a proton pump inhibitor (PPI) and two of the following antibiotics: Clarithromycin, nitro-imidazole or amoxicillin [5]. Unfortunately, as observed with other bacterial pathogens, antibiotic resistance of *H. pylori* increased worldwide leading to its classification by the World Health Organization (WHO) in 2017 as “High priority pathogen”. Thus, rates of resistance to clarithromycin, metronidazole or amoxicillin may be as high as 15–50%, 35–46% and 3–60%, respectively depending on the country. In addition, the prevalence of multiresistant *H. pylori* rose, for instance 20–40% being resistant to two or more antibiotics in Iran in 2015 [6]. Opposite to the case of other infections where eradication therapy results in nearly 100% successful rate, the story is different with *H. pylori* with eradications rates between 70% and 90% in the best case [7]. Another meta-analysis published two years ahead, have shown that the treatment failure rate with the most commonly used first-line therapies, proton pump inhibitors (PPIs) plus two antibiotics, goes up to 20% of patients [8]. The future is getting dark regarding antibiotic’s resistance in bacterial pathogens, including *H. pylori*. The WHO predicts an exponential increase in the number of multi-resistant strains in the near future. Altogether, we are now entering a new era of the resistance which calls for new strategies to deal with these *H. pylori* resistant strains. Apart from numerous therapeutic regimens suggested, only few effective antibiotics were established. An alarming situation that calls for more thoughtful application of current available antibiotics. Hence, new strategy toward the *H. pylori* treatment should be taken into consideration for reaching the 100% expected rate of eradication. Due to all what preceded, concerns towards *H. pylori* nowadays are growing fast. The number of peer-reviewed publications on *H. pylori* has rapidly increased, from less than 200 in 1990 to approximately 1500 per year over the last few years [9]. 

For these reasons new approaches with new antibiotics should be used for the eradication of *H. pylori*. One group of antibiotics which may be promising are the antimicrobial peptides (AMP). Their particular mechanism of action (MOA) relying on their fast insertion into the bacterial membrane leading to its permeabilization makes them: (i) active against bacteria already resistant to classical antibiotic, such as the ones currently used to treat *H. pylori* infection, targeting intracellular enzyme and (ii) nearly impossible to cause resistance [10,11]. Indeed, few studies have already proved the efficiency of some AMP to kill *H. pylori* [12].

Temporins are AMP with known activity against numerous bacteria [13], but their activity on *H. pylori* has never been addressed. Temporin family is one of the largest of AMPs produced by skins of ranid frogs with more than 100 members [13] all linear, hydrophobic and weakly positively charged peptides derived from skin secretions of Eurasian and ranid frog [14,15,16]. They are the shortest peptides in nature with minimal model for membrane destabilizing which makes them an attractive template for new therapeutic agents and antimicrobial agents [17].

Among the various isoforms of temporins, temporin-SHa (Figure 1) has attracted considerable attention due to its interesting properties. Though not extensively studied, temporin-SHa showed promising results in the studies it was implemented in. In a very recent study, temporin-SHa and some analogs were tested, having grafted on some surfaces to produce Biocidal/antiadhesive ones. Results were very interesting with 80% to 90% killing efficiency [18,19]. Another study published in 2019 compared the antiviral activity of temporin-SHa and [K³]SHa to LL-37 and temporin-Tb against Herpes Simplex Virus Type 1. Results were also promising for both temporin-SHa and its analog [K³]SHa (serine3 changes to lysine) that directly acted as anti-HSV-1 peptides [20]. 

Raja et al. in 2017, performed a study on temporin-SHa showing its high potential broad-spectrum antiparasitic and antibacterial agent [21]. 

Recently, new analogs of temporin-SHa (SHa) were synthetized with substitution of Gly residues at positions 4, 7, and/or 10 with D-Ala or L-Ala (Table 1). These analogues showed interesting anticancer activity, particularly temporin[G10a]-SHa (NST1) that was found very potent against difference cancer cell lines [22]. One interesting thing previously shown by Shah et al. is the dramatic increase in the activity of temporin-LK1 when single Gly was substituted by D-Ala [23]. Also, temporin-SHa is well known for being active against bacteria, yeasts, fungi, and protozoa [24]. In the present study the activity of the native temporin-SHa and its analogs were tested against *H. pylori*.

## 2. Materials and Methods 

### 2.1. Synthesis of Temporin-SHa and Its Analogs

Fmoc-protected amino acids, Rink amide resin and coupling reagents were purchased from Novabiochem (Hohenbrunn, Germany) and Chem-impex (Wood Dale, IL, USA). Spectras of 1H (600 MHz) and 13C (125 MHz) were obtained at room temperature from Bruker nuclear magnetic resonance (NMR) spectrometers (Bruker, Fällanden, Uster (district) in canton of Zürich, Switzerland). Herein operating frequency of NMR spectrometers was adjusted at 600 MHz and 125 MHz respectively and solvent was dimethyl sulfoxide (DMSO)-d6 of Sigma Aldrich (St. Louis, MO, USA). Ultraflex III TOF/TOF (BrukerDaltonics, Bremen, Germany) MALDI spectrometer was used to record HRMALDI-MS spectra. Furthermore, purification of peptides was performed through LC-908W RP-HPLC (Japan Analytical Industry, Mizuho, Nishitama Tokyo, Japan). While the column of ODS-MAT 80 (C18) was used. To further check the purity of peptides, ultra-performance liquid chromatography (UPLC) (Agilent 1260 Infinity Diode Array, C-4 reversed-phase analytical column, 5 μm, 150 × 4.6 mm, Santa Clara, CA, USA) was used. Synthesis was performed using the solid phase peptide synthesis (SPPS) method wherein the chain was built on insoluble Rink amide MBHA resin (0.51 mmol/g, Novabiochem) support in a stepwise process. Resin was swelled in DCM and then soaked in DMF for one hour. Afterwards, it was treated with 20% 4-methylpiperidine to remove Fmoc group from resin. Then first amino acid Phenylalanine, five equivalence dissolved in 10 mL DMF, was loaded using five equivalences of OxymaPure (ethyl (hydroxyimino)cyanoacetate)) and five equivalences of *N,N*′-diisopropylcarbodiimide (DIC). Then, the reaction mixture was agitated for 24 h for maximum coupling of first amino acid. This process was repeated again in sequence of deprotection by 4-methyl piperidine and then loading of successive amino acids dissolved in 10 mL of DMF along with OxymaPure and DIC. Furthermore, the ninhydrin colorimetric test was performed to check the completion of reaction. The sequence was completed with the removal of Fmoc group of last amino acid and then the cleavage was performed using cocktail of 95% trifluoro acetic acid (TFA), 1% triisopropylsilane (TIPS), 2% deionized water (D.I) and 2% ethanedithiol (EDT). This was followed by precipitation of crude product using diethyl ether and centrifugation. Further purification was executed by recycling preparative -HPLC. Meanwhile, the solvent system was maintained at 60% acetonitrile (ACN), 40% de-ionized water (D.I) and 0.08% trifluoro acetic acid (TFA) at 2 flow rate.

### 2.2. Mass Spectrometric Analysis of Peptides

For mass determination Matrix-assisted laser desorption ionization (MALDI) was carried out on an ultraflex III TOF/TOF (Bruker Daltonics, Bremen, Germany) mass spectrometer. To perform this, crude peptide was dissolved in water and acetonitrile mixture 40:60 and TFA was used in 0.08%. This solvent system was deposited on MALDI plate and spectra was recorded. In the same way, electrospray ionization quadrupole time-of-flight mass spectrometry (ESI-QTOF-MS) spectra was acquired on Q-STAR XL mass spectrometer. Full scan of each spectra was obtained at flow rate of 3 μL/min to acquire full scan. Afterwards, spectra were directly infused into mass spectrometer. A 5500 V of electrospray voltage at spraying needle was used for the optimization.

### 2.3. Peptides Specifications

The natural peptide temporin-SHa and its analogs were synthesized by using standard protocol of Fmoc peptide synthesis on Ring amide AM resin (Table 2). The pure peptide products were obtained by purification on RP-HPLC using the isocratic solvent system. The peptides were charaterized by matrix-assisted laser desorption/ ionization-time of flight (MALDI-TOF) mass spectrometry, HR-MALDITOF-MS, MALDI/MS/MS and NMR studies [22].

Temporin SHa: (H-Phe^1^-Leu^2^-Ser^3^-Gly^4^-Ile^5^-Val^6^-Gly^7^-Met^8^-Leu^9^-Gly^10^-Lys^11^-Leu^12^-Phe^13^-NH_2_) retention time (tR) 4.638 min, HR-MALDI *m*/*z* 1402.7896 [M + Na]^+^ (C_67_H_109_N_15_NaO_14_S calc. for 1402.7891).

Temporin [G10a]SHa (NST1): (H-Phe^1^-Leu^2^-Ser^3^-Gly^4^-Ile^5^-Val^6^-Gly^7^-Met^8^-Leu^9^-D-Ala^10^-Lys^11^-Leu^12^-Phe^13^-NH_2_). [G10a] SHa was synthesized by substitution of Gly^10^ residue of natural product peptide SHa by D-Ala. tR 3.489 min, HR-MALDI *m*/*z* 1394.822 [M+H] ^+^ (C_68_H_112_N_15_O_14_S calc. for 1394.823).

Temporin[G4a]SHa (NST2): (H-Phe^1^-Leu^2^-Ser^3^-D-Ala^4^-Ile^5^-Val^6^-Gly^7^-Met^8^-Leu^9^-Gly^10^-Lys^11^-Leu^12^-Phe^13^-NH_2_). Peptide [G4a] SHa was synthesized by substitution of Gly4 residue of natural product peptide SHa by D-Ala. tR 3.673 min, HR-MALDI *m*/*z* 1416.8048 [M + Na] + (C_68_H_111_N_15_NaO_14_S calc. for 1416.8053).

Temporin[G7a]SHa (NST3): (H-Phe^1^-Leu^2^-Ser^3^-Gly^4^-Ile^5^-Val^6^-D-Ala^7^-Met^8^-Leu^9^-Gly^10^-Lys^11^-Leu^12^-Phe^13^-NH_2_). [G7a] SHa was synthesized by substitution of Gly^7^ residue of natural product peptide SHa with D-Ala. tR 3.381 min, HR-MALDI *m*/*z* 1416.8084 [M + Na]^+^ (C_68_H_111_N_15_NaO_14_S calc. for 1416.8053).

Temporin[G4,7a]SHa (NST4): (H-Phe^1^-Leu^2^-Ser^3^-D-Ala^4^-Ile^5^-Val^6^-D-Ala^7^-Met^8^-Leu^9^-Gly^10^-Lys^11^-Leu^12^-Phe^13^-NH_2_). [G4,7a] SHa was synthesized by substitution of Gly^4^ and Gly^7^ residues of natural product peptide SHa by D-Ala. tR 3.467 min, HR-MALDI *m*/*z* 1430.8204 [M + Na] + (C_69_H_113_N_15_NaO_14_S calc. for 1430.8204).

Temporin[G4,7,10a]SHa (NST5): (H-Phe^1^-Leu^2^-Ser^3^-D-Ala^4^-Ile^5^-Val^6^-D-Ala^7^-Met^8^-Leu^9^-D-Ala^10^-Lys^11^-Leu^12^-Phe^13^-NH_2_) [G4,7,10a] SHa was synthesized by substitution of all three glycine residues of peptide with D-Ala. tR 3.501 min, HR-MALDI *m*/*z* 1444.8361 [M + Na]^+^ (C_70_H_115_N_15_NaO_14_S calc. for 1444.8366).

Temporin[G7,10a]SHa (NST6): (H-Phe^1^-Leu^2^-Ser^3^-Gly^4^-Ile^5^-Val^6^-D-Ala^7^-Met^8^-Leu^9^-D-Ala^10^-Lys^11^-Leu^12^-Phe^13^-NH_2_) [G7,10a] SHa was synthesized by substitution of Gly^7^ and Gly^10^ residue of natural product peptide by D-Ala. tR 3.361 min, HR-MALDI *m*/*z* 1430.8209 [M + Na]^+^ (C_69_H_113_N_15_NaO_14_S calc. for 1430.8204).

Temporin[G4,10a]SHa (NST7): (H-Phe^1^-Leu^2^-Ser^3^-D-Ala^4^-Ile^5^-Val^6^-Gly^7^-Met^8^-Leu^9^-D-Ala^10^-Lys^11^-Leu^12^-Phe^13^-NH_2_) [G4,10a] SHa was synthesized by substitution of Gly^4^ and Gly^10^ residue of natural product peptide by D-Ala. tR 3.052 min, MALDI *m*/*z* 1430.8 [M + Na]^+^ (C_69_H_113_N_15_NaO_14_S calc. for 1430.8204).

Temporin[G4,10A]SHa (NSTL7): (H-Phe^1^-Leu^2^-Ser^3^-L-Ala^4^-Ile^5^-Val^6^-Gly^7^-Met^8^-Leu^9^-L-Ala^10^-Lys^11^-Leu^12^-Phe^13^-NH_2_) [G4,10A] SHa was synthesized by substitution of Gly^4^ and Gly^10^ residue of natural product peptide by L-Ala. tR 4.173 min, MALDI *m*/*z* 1430.8 [M + Na]^+^ (C_69_H_113_N_15_NaO_14_S calc. for 1430.8204).

### 2.4. Microorganism Strains and Growth Conditions

*H. pylori* used in this study were either the reference strain obtained from the American Type Culture Collection (ATCC 43504) or clinical strains received frozen from our collaborator Josette Raymond from Hospital Cochin, Paris. Strains were obtained from patients who underwent a gastroendoscopy performed for various gastro-intestinal disorders according to the recommendations of Haute Autorité de Santé (HAS) [25]. During the endoscopy, biopsy samples were taken from antrum and corpus. Samples were then crushed, and the material was used for bacterial culture and molecular methods. An antibiogram was performed and strains were then frozen at -80 °C in a broth with 10% glycerol. Bacteria were cultured as previously described [26]. Briefly, *H. pylori* (ATCC 43504) was cultured in brain heart infusion (BHI) and incubated at 37 °C for 16 h in microaerophilic condition in BD GasPak EZ using BD GasPak EZ CampyPak container system sachets (BD Biosciences, Le Pont-de-Claix, France). For clinical strains, a sample from the frozen bacteria was streaked on Blood Agar (Base) (Sigma-Aldrich, Illkirch-Graffenstaden, France) supplemented with 5% horse defibrinated blood (Fisher Scientific, Les Ulis, France) and 2% isovitale-X (Fisher Scientific, Illkirch-Graffenstaden, France). The petri dishes were then incubated at 37 °C for 4–7 days in microaerophilic condition. Then, a colony is transferred to 3 mL Mueller Hinton (MH) broth supplemented with 10% fetal bovine serum (FBS) and 2% isovitale-X. The cultures were then incubated for 4–7 days until the bacteria grow before further experiments.

### 2.5. Antimicrobial Activity Assay

Antimicrobial activity of temporin-SHa and its analogs was evaluated using determination of the minimal inhibitory concentration (MIC) using two-fold serial dilutions of antimicrobial peptides in bacterial liquid media following the National Committee of Clinical Laboratory Standards (NCCLS, 1997) as previously described [10,26]. Briefly, after the growth of *H. pylori* in liquid culture the optical density (OD) of the bacterial suspensions were read at 600 nm and adjusted to one. Stock solutions of the peptides at 1 mM were prepared in sterile water. Bacteria were diluted in medium to reach bacterial density around 10^5^ bacteria/mL. One-hundred microliters per well of bacterial suspension were then added into sterile polypropylene 96-well microplates (Greiner BioOne, Paris, France) and exposed to increasing concentrations of peptides (from zero to 100 µM, 1:2 serial dilution). Media used for *H. pylori* (ATCC 43504) in MIC assay was MH, whereas for the clinical strain MH broth supplemented with 10% FBS and 2% isovitale-X was used. Volume of water corresponding to the highest dose of peptides tested was used as negative control and was found inactive. Plates were incubated at 37 °C for 24 h for *H. pylori* (ATCC 43504) and for 4–7 days for the clinical strains. *H. pylori* was tested in microaerophilic conditions generated using micro-anaerobic BD GasPak system. At the end of the incubation, OD_600nm_ was measured using microplate reader (Synergy Mx, Biotek, Colmar, France). The MIC was defined as the lowest concentration of drug that inhibited visible growth of the organism. Experiments were conducted in independent triplicate (*n* = 3).

### 2.6. Effect of the Stomach Conditions on Antibacterial Effect of Temporin-SHa and Its Analogs

The stomach is an acidic environment (pH around 2) also containing pepsin that is able to digest proteins and peptides. It is also rich with mucus to protect it from the aggressive acidity. For this reason, the impact of mucins and pepsin on the activity of the peptides was investigated. The impact of the gastric mucins on peptide activity was evaluated using a combination of commercial porcine stomach mucins MUC2 and MUC3 (Sigma–Aldrich) at a final concentration of 1 mg/mL sterilized using 0.2 µm filter. This concentration was used based on the estimated concentration of mucins present in the mucus in vivo [27]. The effect of mucins on peptides activity was evaluated by MIC and by time killing assay as explained below. The resistance of the peptides to pepsin was tested as previously described with some changes [28]. Briefly, temporin-SHa and its analogs (at 1 mg/mL, final concentration) were incubated for 4 h at 37 °C with 1 mg/mL (final concentration) of pepsin from porcine gastric mucosa (Sigma-Aldrich) in HCl 10mM, pH 2. This concentration was used since it corresponds the pepsin concentration in the stomach (0.5–1 mg/mL) [29]. At the end of the digestion, the activity of the temporin-SHa and its analogs on *H. pylori* was evaluated by MIC measurement, digested peptides being diluted in bacterial media, resulting in pepsin inhibition due to pH effect, pepsin being only active at acidic pH. Pepsin alone was used as negative control and showed no activity against *H. pylori* in this condition.

### 2.7. Time Kill Assay

The bactericidal activity of peptides was assessed using the time kill assay as previously described with some modifications [30]. Briefly, overnight liquid culture of *H. pylori* (ATCC 43504) was prepared and an optical density measurement (OD_600nm_) was taken to provide an estimation of cell numbers in the culture. Bacterial suspension was then centrifuged at 3000 rpm for 5 min and the bacterial pellet was then resuspended in sterile PBS pH6 at approximatively 10^7^ CFU/mL. The pH of PBS was changed from 7 to 6 with HCl to mimic the micro-environment surrounding *H. pylori* in the stomach. Indeed, although the lumen of the stomach is around 1–2, *H. pylori* is able, through the activity of its urease, to create a local micro-environment around it of pH 6 [31]. In addition to PBS pH 6, the time kill assay was also performed in PBS pH 6 containing a mixture of porcine stomach mucins MUC2 and MUC3 (at 1 mg/mL, final concentration) or in the presence of human gastric cells N87 producing human mucus. In that particular case, N87 cells were seeded onto 96-wells plates and left to grow for 48–72 h until they reached confluence. At that time, wells were aspirated and bacterial suspension in PBS pH 6 were added to the wells. In all cases, resuspended bacteria (200 µL) were treated with peptides or the conventional antibiotic amoxicillin at a final concentration corresponding to five times their MIC. Bacteria were then incubated in micro-aerobic conditions at 37 °C using GasPak units. After five, 20, 60 or 180 min, bacterial suspensions were serially diluted 1:10 in PBS before 10 µL from each dilution were to LB agar petri dish and streaked. The petri dishes were then incubated 16 h at 37 °C in microaerophilic conditions before counting of growing colonies. The number of surviving bacteria was calculated using the number of observed colonies and the dilution factor. The experiment was done in triplicate.

### 2.8. Membrane Permeabilization Assay

Bacterial membrane permeabilization was evaluated using the cell-impermeable DNA/RNA probe propidium iodide as previously explained [26,32,33]. Logarithmic growing bacterial suspension of *H. pylori* (ATCC 43504) was prepared from overnight bacterial suspension after 1:10 dilution. After 3 h incubation at 37 °C, in microaerophilic condition, bacterial suspensions were centrifuged for 5 min at 3000× *g*. Cell pellets were then resuspended in sterile PBS at about 109 bacteria/mL. Propidium iodide (solution at 1 mg/mL, Sigma Aldrich) was then added to the suspension at a final concentration of 60 µM. 100 µL of this suspension were then transferred into 96-well black plates and were exposed to increasing concentrations of peptides obtained by serial dilution (from zero to 100 µM, 1:2 dilution). The volume of water (vehicle) corresponding to the highest doses of peptides was used as negative control and was found inactive. Kinetics of fluorescence variations (excitation at 530 nm and emission at 590 nm) were then recorded over 240 min of incubation at 37 °C in micro-aerobic condition using a microplate reader (Biotek, Synergy Mx). The effect of the peptides on the membrane integrity was expressed in percentage of permeabilization, CTAB at 300 µM being used as positive control giving 100% permeabilization. All experiments were done in triplicate.

### 2.9. Lipid Insertion Assay

Peptide–lipid interaction was measured using reconstituted lipid monolayer at the air-water interface as previously described [32,33]. Total lipids were extracted from overnight cultures of *H. pylori* (ATCC 43504) using Folch extraction procedure as previously described [32,33]. Extracted total lipids were dried, resolubilized in chloroform:methanol (2:1, *v/v*) and stored at –20 °C under nitrogen. For peptide-lipid interaction assay, total lipid extract was spread using a 50 µL Hamilton’s syringe at the surface of 800 µL of sterile PBS creating a lipid monolayer at the air-water interface. Lipids were added until the surface pressure reached the desired value. After 5–10 min of incubation allowing evaporation of the solvent and stabilization of the initial surface pressure, 8 µL of peptide diluted in sterile PBS at 100 µM were injected into the 800 µL sub-phase of PBS under the lipid monolayer (pH 7.4, volume 800 µL) using a 10 µL Hamilton syringe giving a final concentration of peptides of 1 µM, preliminary experiments having shown that this concentration was optimal. The variation of the surface pressure caused by peptide injection was then continuously monitored using a fully automated microtensiometer (µTROUGH SX, Kibron Inc., Helsinki, Finland) until reaching equilibrium (maximal surface pressure increase being usually obtained after 15–25 min). The critical pressure of insertion of temporin-SHa and its analogs in the total lipid extract from *H. pylori* was determined as previously described [33] by changing the initial pressure of lipid monolayer (from 10 and 30 mN/m) and measuring the variation of pressure caused by the injection of peptide. All experiments were carried out in a controlled atmosphere at 20 °C ± 1 °C and data were analyzed using the Filmware 2.5 program (Kibron Inc.). Variation of surface pressure was plotted as a function of initial surface pressure and the critical pressure of insertion of each peptide into *H. pylori* lipids was calculated as the theoretical value of initial pressure of lipid monolayer not permissive to peptide insertion, i.e., giving a variation of pressure equal to 0 mN/m. The accuracy of the system under our experimental conditions was determined to be ± 0.25 mN/m for surface pressure measurements.

### 2.10. In Vitro Toxicity Assay Using Human Gastric Cells

The impact of temporin-SHa and its analogs on the viability of human gastric cells was evaluated using resazurin-based assay with some modifications. [26] Human gastric cells N87 (ATCC CRL-5822) were cultured in RPMI-1640 medium supplemented with 10% fetal calf serum (FCS), 1% L-glutamine and 1% antibiotics (all from Invitrogen, Carlsbad, CA, USA) and were routinely seeded onto 25 cm^2^ flasks maintained in a 5% CO_2_ incubator at 37 °C. For the toxicity assay, cells grown on 25 cm^2^ flasks were detached using trypsin–EDTA solution (from Thermofisher), counted using Mallasez counting chamber and seeded into 96-well cell culture plates (Greiner Bio-one) at approximately 104 cells per well. The cells were left to grow for 48–72 h at 37 °C in a 5% CO_2_ incubator until they reached confluence. Plates were then aspirated and increasing concentrations of peptides were added to the cells for 48 h at 37 °C in a 5% CO_2_ incubator, volume of water corresponding to the volume of peptides being used as negative control. At the end of the incubation, wells were aspirated, and cell viability was evaluated using resazurin-based in vitro toxicity assay kit (Sigma-Aldrich) following manufacturer’s instructions. Briefly, resazurin stock solution was diluted 1:10 in sterile PBS containing calcium and magnesium (PBS^++^, pH 7.4) and empty wells were filled with 100 µL of the diluted solution. After 4 h incubation at 37 °C, fluorescence intensity was measured using microplate reader (Synergy Mx, Biotek) with an excitation wavelength of 530 nm and an emission wavelength of 590 nm. The fluorescence values were normalized by the controls and expressed as percent viability. The IC_50_ (the inhibitory concentrations reducing 50% of the cell viability) values of peptides on cell viability (i.e., the concentration of peptides causing a reduction of 50% of the cell viability) were calculated using the GraphPad^®^ Prism 7 software (San Diego, CA, USA).

### 2.11. Ex-Vivo Toxicity Assay Using Human Gastric Explants

To further evaluate the innocuity of the peptides, ex vivo experiments were performed using human explants as previously described [34]. Briefly, human gastric tissue samples were obtained from patients undergoing surgery at the Hospital Laveran (Marseille, France), unit of gastrointestinal surgery according to a collaborative clinical transfer project. The procedures were approved by the French ethic committee (CODECOH n° DC-2019-3402). All patients agreed for the use of their tissues for research purposes. Samples were taken from healthy patients during sleeve gastrectomy from macroscopically unaffected area as identified by the surgeons. After resection, the specimens were placed in ice-cold oxygenated sterile DMEM solution containing 1% streptomycin/penicillin solution and 50 μg/mL gentamycin and were directly transferred to the laboratory within 15 min on ice. Gastric tissues were extensively washed and maintained in ice-cold culture media containing 1% streptomycin/penicillin solution and 50 μg/mL gentamycin. Tissues were cleaned from vascular vessels and conjunctive tissue using forceps under binocular microscope. Gastric explants (wet weight of 20–30 mg) were then isolated from the cleaned resections using surgical punch (diameter of 0.5 cm^2^). All these operations were achieved in less than 2 h after the resections were obtained from the surgery unit. Finally, the explants were washed three times with culture media without antibiotics and transferred into 24-well plates before being incubated in RPMI media without fetal bovine serum, antibiotics and phenol red at 37 °C in the 5% CO_2_ incubator with 100 µM of peptides for 2, 4 or 8 h. Phenol red was omitted from the culture media since it interferes with the lactate dehydrogenase (LDH) assay. The detergent cetyl-trimethylammonium bromide (CTAB) at 300 µM was used as positive control of tissue damages. At the end of the incubation, the explants were collected and washed three times with PBS^++^ and were fixed overnight at 4 °C with PFA diluted at 4% in PBS. After that, gastric explants were washed twice with PBS and were included in an inclusion medium (TFM-EMS), in transverse position to allow cutting in the crypt-villosity axis using cryostat (Leica CM3050) (Leica Microsystems, Nanterre, France). Four sections of 5 µm thickness were obtained per explant, each section being separated by 100 µm from the next in order to cover all the tissue. Explants were then stained using hematoxylin and eosin (H&E) staining protocol. Briefly, samples were incubated for 8 min in hematoxylin (Sigma-Aldrich) and then allowed to stain by incubation with water for 2 min. Then explants were incubated for 1 min in eosin (Sigma-Aldrich) and then in water for 1 min. This was followed by incubation of explants with ethanol at concentration of 70% then 95% and finally 100% for 2 min each. After blotting excess ethanol, gastric explants were incubated for 15 min with xylene and mounted with coverslip using Eukitt mounting media (EMS, Hatfield, PA, USA). Finally, explants were left overnight to dry before examination of tissue organization under the microscope (Leitz DMRB microscope (Leïca), equipped with Leïca DFC 450C camera). Quantification of the release of LDH was also used in addition to microscopic observations to evaluate the impact of peptides on human gastric explants. Briefly, after 2, 4 or 8 h of incubation of the explants with the peptides, the supernatants of the wells were collected and stored at -80 °C before the LDH assay. The LDH activity was measured using previously described protocol with minor changes [35]. Briefly, 50 µL of supernatants obtained from explants were mixed with 200 µL of Tris-Lactate buffer (containing Tris HCl at 86 mM, L-Lactate at 56 mM and KCl at 172 mM, pH 9.3, all from Sigma-Aldrich) and 20 µL of NAD (at 172 mM, Sigma-Aldrich) into well of 96-wells plates. Increase in OD_339nm_ was then measured at 30 °C after 0, 1, 5, and 10 min using microplate reader (Synergy Mx, Biotek). The release of intracellular LDH caused by the peptides was expressed as a percentage of total release, CTAB was used as positive control giving 100 % release of LDH. The experiment was performed in triplicate.

## 3. Results

### 3.1. Antibacterial Activity of Temporin-SHa and Its Analogs against H. pylori

Antimicrobial activity of temporin-SHa and its analogs was first evaluated using the reference strain *H. pylori* (ATCC 43504) using the broth microdilution method (Table 3). Results showed that all the peptides were active having different potencies. For instance, NST1 had the strongest activity with an MIC of 1.5 µM followed by temporin-SHa and NST3, NST6, and NST7 which share the same MIC of 3.12 µM. NST2 and NST4 had similar potencies with MIC values of 12.5 and 6.25–12.5 µM, respectively. Finally, the less active peptides were NST5 and NSTL7 with MIC values of 100 and 50 µM respectively.

### 3.2. Evaluation of the Pore-Forming Activity of Temporin-SHa and Its Analogs on H. pylori

Evaluation of the impact of temporin-SHa and its analogs on the membrane integrity of *H. pylori* (ATCC 43504) was done using the cell impermeable DNA probe propidium iodide. Dose- and time-dependent effects were evaluated (Figure 2 and Figure 3). Results showed that all the peptides were able to permeabilize the membrane of *H. pylori* in a concentration-dependent manner. The speed of permeabilization was fast, the maximum effect being seen within the first five minutes. Differences were found between the peptides, in accordance with their MIC values, confirming that the mechanism of action of temporin-SHa and its analogs on *H. pylori* relies on pore-forming activity. Thus, the strongest permeabilization was found with NST1 followed by NST3, temporin-SHa, NST6, NST7, NST4, and NST2. Finally, the peptides with the weakest permeabilization effect were NST5 and NSTL7. Pore-forming activity was thus in accordance with the MIC of the peptides, i.e., 1.5, 3.12, 3.12, 3.12, 3.12, 6.25 -12.5, 12.5, 50, and 100 µM, for NST1, temporin-SHa, NST3, NST6, NST7, NST4, NST2, NST7, NST5, respectively).

### 3.3. Evaluation of the Interaction of Temporin-SHa and Its Analogs with Lipids

After knowing that the temporin-SHa and its analogs are able to form pores in the membrane of *H. pylori*, their interaction with total bacterial lipids was evaluated. First, the insertion of the peptides in monolayers formed by total lipid extracts of *H. pylori* (ATCC 43504) was evaluated by measuring the critical pressure of insertion as explained in Materials and Methods. The variation of surface pressure caused by the insertion of temporin-SHa and its analogs was plotted as a function of initial surface pressure (Figure 4). The critical pressure of insertion of each cyclic peptide in total lipid extract of *H. pylori* was calculated as the theoretical value of initial pressure of lipid monolayer not permissive to peptide insertion, i.e., causing a variation of pressure equal to 0 mN/m. Critical pressure of insertion is a very important parameter as it reflects the ability of a peptide to insert into the lipid monolayer, its value increasing for peptide with higher insertion capacity. Results showed that the critical pressures of insertion of temporin-SHa and its analogs had some difference between each other (Table 4). Interestingly, values of critical pressure of insertion reflected the MIC results obtained with the peptides since the highest critical pressure of insertion were found for the peptides with the lowest MIC, except for NST7. For example, NST6 (44.43 mN/m), SHa (39.66 mN/m), NST4 (39.12 mN/m), NST1 (38.65 mN/m), NST2 (37.86 mN/m), and NST3 (35.73 mN/m) had the highest critical pressure of insertion, agreeing with their antibacterial activity against *H. pylori* ranging from 1.5 to 12.5 µM. The critical pressure of insertion of NST5, NST7, and NST7L were found the lowest with a value of 33.99 and 33 mN/m for NST5 and NST7, respectively, NSTL7 being even not able to insert at all in total lipids of *H. pylori*. In the case of NST5 and NST7L, this result was in accordance with their poor antibacterial activity against *H. pylori* (MIC of 100 and 50 µM for NST5 and NST7L, respectively). What was surprising is that, although it has a poor ability to insert into monolayer of total extract of *H. pylori*, NST7 has a strong antibacterial activity against *H. pylori* (Table 3) and is able to permeabilize its membrane (Figure 2) suggesting a different permeabilization mechanism in that case compared to temporin-SHa and the other analogs.

### 3.4. In Vitro Evaluation of the Toxicity of Temporin-SHa and Its Analogs on Human Gastric Cells

Antimicrobial efficiency is not sufficient to develop a therapeutic molecule, it must also be safe for the host. Therefore, cytotoxicity of temporin-SHa and its analogs was assayed on N87 cells, a model of human gastric cells. Except for temporin-SHa and its analogs NST2 and NSTL7, all the analogs displayed a dose-dependent reduction of human gastric cell viability after 48 h of incubation (Figure 5). Theoretical IC_50_ on cell viability (i.e., the inhibitory concentrations reducing 50% of the cell viability) of each peptide were calculated using GraphPad^®^ Prism 7 (Table 3). Results showed that the order of toxicity of the analogs when comparing the IC_50_ is as follows. The less toxic was NSTL7 with an IC_50_ of 2290 ± 762 µM followed by temporin-SHa > NST2 > NST5 > NST3 >NST4 > NST6 > NST1 with IC_50_ of 1558 ± 324, 1247 ± 240, 762.1 ± 235, 520.1 ± 148, 306.5 ± 48.4, 104.4 ± 9.3, and 90.41 ± 10.4 µM, respectively. The most toxic peptide was NST7 with an IC_50_ of 84.69 ± 9.0 µM. The safety of the peptides was further evaluated by calculating the therapeutic index (TI) of each analog. Therapeutic index was calculated by dividing the IC_50_ of each peptide by its MIC against *H. pylori* (ATCC 43504) (Table 3). Results showed that on one hand temporin-SHa and analogs NST3, NST2, and NST1 had the widest therapeutic index with values of 499.35, 166.6, 99.76, and 60.27, respectively. While on the other hand NSTL7, NST6, NST7, NST4, and NST5 showed the narrowest therapeutic index with values of 45.8, 33.46, 27.14, 24.5, and 7.62, respectively. Following this step, the safest peptides with the widest therapeutic index (i.e., temporin-SHa and its analogs NST1, NST2 and NST3) were chosen to continue their characterization (Figure 6).

### 3.5. Antibacterial Activity of Temporin-SHa and Its Analogs on Clinical Strains of H. pylori

The activity of temporin-SHa and its analogs NST1, NST2, and NST3 was tested on clinical strains of *H. pylori* obtained from patients and resistant to conventional antibiotics such as clarithromycin and metronidazole (Table 5, Figure 7). Results showed that the four peptides were active against all resistant clinical strains with temporin-SHa, NST1, and NST3 being the more active with MIC values of 12.5–25, 6.25–12.5, 50–100, and 12.5–25 µM for temporin-SHa, NST1, NST2, and NST3, respectively. Clinical strains are fastidious to grow, requiring the addition of 10% FBS to MH media. To know if the higher MIC found on clinical strains compared to the ATCC strain were due to the presence of FBS in the media, MIC of the four peptides were repeated on the reference *H. pylori* strain (ATCC 43504) in the same media. In MH supplemented with 10% FBS, MIC of temporin-SHa, NST1, NST2, and NST3 on *H. pylori* (ATCC 43504) were found higher than in MH media (with MIC values of 3.12, 1.5, 12.5, and 3.12 µM versus 12.5, 6.25, 50, and 12.5 µM, in MH and MH supplemented with 10% FBS, respectively). This showed that the slight decrease in antibacterial potency of the peptides observed with the clinical strains was not due to a lower sensitivity of these strains to the peptides but to the presence of FBS in the assay, suggesting that the MIC obtained with clinical strains were in fact over-estimated and will be lower in vivo in the stomach environment with no FBS.

### 3.6. Evaluation of the Impact of the Physiologic Stomach Conditions on the Activity of the Temporin-SHa and Its Analogs

The stomach environment is characterized by an acidic pH and the presence of mucins forming the mucus and protease, i.e., pepsin. The effect of stomach conditions on the activity of temporin-SHa and its analogs was evaluated as explained in Materials and Methods. In the presence of the gastric mucins MUC2 and MUC3 at 1 mg/mL, a concentration found in vivo [27], the MIC of temporin-SHa, NST1, NST2, and NST3 was slightly increased by two-fold with 3.12, 1.5, 12.5, and 3.12 µM and 6.25, 3.12, 25, and 6.25 µM in the absence and presence of mucins, respectively (Table 6). After 4 h incubation of temporin-SHa, NST1, NST2, and NST3 at 37 °C with a physiological concentration of pepsin in HCl pH 2 (i.e., 1 mg/mL) [29], only analog NST1 retained its activity (MIC = 25 µM) and hence possessed partial resistance towards pepsin (Table 6). All the other peptides were seen inactive with an MIC > 50 µM. After knowing that analog NST1 was the only analog resistant to pepsin, it was chosen for other assays done in this study, temporin-SHa being used as a comparative control.

### 3.7. Time Kill Assay

The kinetic of killing of *H. pylori* by temporin-SHa and NST1 was studied as explained in Materials and Methods. The reference strain *H. pylori* (ATCC 43504) was incubated with either temporin-SHa, its analog NST1 or the conventional antibiotic amoxicillin at 5-times their MIC followed by counting of the colony forming units (CFU) remaining alive at different time intervals. Analysis was first performed in PBS pH 6, this pH being selected due to the fact that, although pH of the stomach is around 1-2, *H. pylori* is able to create through the activity of its urease a local environment around it at pH 6 [31]. Results showed that temporin-SHa and NST1 have a rapid bactericidal activity in comparison to amoxicillin. Thus, after 20 min of incubation amoxicillin, temporin-SHa and NST1 decreased the number of CFUs by around 0.8 log (9% killing), two Logs (87% killing) and three Logs (99% killing), respectively (Figure 8A,B). After 3 h of incubation, NST1 totally killed *H. pylori* with 0% survival, temporin-SHa and amoxicillin causing 95 and 34% killing, respectively. The killing assay was also performed in the presence of porcine gastric mucins (MUC2 and MUC3 at 1 mg/mL) (Figure 8C,D) or in the presence of the human gastric cells N87 producing human gastric mucus (Figure 8E,F). Results showed that the presence of mucus slowed down the killing activity of the peptides, temporin-SHa and NST1 needing 60 min to reach 86–95% and 91–98 % killing, respectively. Although the killing took longer, both peptides were able to kill more than 99.6% of *H. pylori* after 3 h of incubation. Amoxicillin gave similar results in the absence or presence of mucus with less than 31.5–55 % killing after 3 h and 75% killing after 9 h.

### 3.8. Ex Vivo Evaluation of the Inocuity of Temporin-SHa and NST1 Using Human Gastric Explants

Human gastric tissues obtained from healthy donors were used to further evaluate the safety of temporin-SHa and NST1 as described in Materials and Methods. Human gastric explants were exposed to temporin-SHa or NST1 at 100 µM for 2, 4, and 8 h. The detergent CTAB (300 µM) was used as a positive control of tissue damages. The impact of peptides and CTAB on tissue viability and integrity was evaluated by measuring the release of LDH and by microscopic observations, respectively. The LDH is an intracellular enzyme, its release is used to evaluate the toxicity of molecules to cells and/or tissue [35]. The quantification of the LDH release (Figure 9) showed that, as negative control, temporin-SHa and its analog NST1 caused minor LDH release compared to the positive control CTAB, with 4.6%, 5.3%, 8.3%, and 81.1%; 8.6%, 10.4%, 12.0%, and 90.3% or 10.7%, 16.8%, 25.0%, and 100% release after 2, 4 or 8 h for negative control, temporin-SHa, NST1 and CTAB, respectively. Interestingly, as found during the in vitro toxicity assay, NST1 showed higher toxicity to human gastric explants compared to temporin-SHa. Accordingly, microscopic observations confirmed that human gastric explants treated for 2, 4 or 8 h with 100 µM of temporin-SHa or NST1 displayed tissue organization close to the negative controls, unlike CTAB that caused major alterations of the tissue architecture visible even after 2 h of incubation (Figure 10 and Figure 11).

## 4. Discussion

In this study temporin-SHa was selected due to its well-known general antimicrobial activity [13,21] to be studied against *H. pylori*. This bacterium is the second cause of bacterial infection with 50% of the World human population being infected (more than 90% in developing countries) and leading to gastric ulcers and gastric cancers for 10% and 1% of patients, respectively. In addition to native peptide, eight analogues of temporin-SHa were tested. They were produced by substitution of one or more of the stereochemically more flexible achiral Gly found on the positions 4, 7, and 10 by D-Ala or L-Ala. These analogs were previously shown to possess anticancer properties, [G10a]SHa (NST1) containing D-Ala in place of Gly at position 10 was the more active on cancer cells [22,23]. The antibacterial activity of temporin-SHa and its analogs against *H. pylori* was first evaluated using MIC determination on the reference ATCC strain. All peptides showed activity with differences in potency. After looking at the sequences of the peptides and comparing them along with the potencies, the peptides could be divided in two groups. First group contains peptides with MIC ranging from 1.5 to 12.5 µM, i.e., temporin-SHa, NST1, NST2, NST3, NST4, NST6, and NST7. The second group contains peptides with MIC ranging from 50 to 100 µM, i.e., NST5 and NSTL7. Using mechanistic approaches, we were able to show that the ability of temporin-SHa and its analogs to insert into lipids from *H. pylori* and to permeabilize its membrane reflects the order of efficiency found in the MIC assay, confirming previous observations obtained by others [21] showing that the antibacterial effect of temporin-SHa and its analogs rely on their insertion into lipids and pore-forming activity.

Following MIC screening, cytotoxicity analysis was done using a human gastric cell model, i.e., N87 cells. Similar to antibacterial screening, results showed different levels of toxicity between the peptides with IC_50_ on cell viability of 1558 ± 324 µM for temporin-SHa and ranging from 84.69 ± 9.0 to 2290 ± 762 µM for its analogs (Table 2). After calculating the therapeutic index or TI (a safety factor calculated by dividing IC_50_ on cell viability by the MIC value of each peptide), two groups can be deduced. One group containing peptides with a TI ranging from 60.27 to 499.35 (including temporin-SHa, NST1, NST2, and NST3) and another group containing peptides with TI ranging from 7.62 to 45.8 (including NST4, NST5, NST6, NST7, and NST7L).

When comparing the sequence of the peptides, it could be observed that Gly at position 4, 7 and 10 play a role in the antibacterial activity and toxicity of the peptides. Thus, NST5 having all Gly replaced by D-Ala lost almost totally its activity against *H. pylori* compared to temporin-SHa (i.e., MIC values of 3.12 and 100 µM for temporin-SHa and NST5, respectively). In the meantime, it was found twice as more toxic to human gastric cells (IC_50_ on N87 cells of 1558 ± 324 and 762.1 ± 235 µM for temporin-SHa and NST5, respectively). Evaluation of the importance of each Gly was then performed by substitution of only one of the three Gly. Results showed that substitution of Gly by D-Ala at position 10 in NST1 increased slightly the activity against *H. pylori* (MIC values of 3.12 and 1.5 µM for temporin-SHa and NST1, respectively) but increased strongly its toxicity to human gastric cells (IC_50_ on N87 cells of 1558 ± 324 and 90.4 ± 10.4 µM for temporin-SHa and NST1, respectively) showing that although Gly10 plays a minor role in the antibacterial activity against *H. pylori*, it is important in term of toxicity, resulting in approximatively a 17-folds increase in the toxicity of the peptide against human gastric cells. Similarly, NST3 with D-Ala substitution at position 7, although saving its activity against *H. pylori* (MIC values of 3.12 µM for temporin-SHa and NST3), displayed an increase (around 3-folds increase) in its toxic effect on human gastric cells (IC_50_ on N87 cells of 1558 ± 324 and 520.1 ± 148 µM for temporin-SHa and NST3, respectively), suggesting that like Gly10, Gly7 is important in term of toxicity of the peptide to human cells. On another hand, NST2 with D-Ala substitution at position 4 displayed a 4-times reduction in the antibacterial activity (MIC values of 3.12 and 12.5 µM for temporin-SHa and NST2, respectively) with a minor change in toxicity, i.e., around 1.2-fold (IC_50_ on N87 cells of 1558 ± 324 and 1247 ± 240 µM for temporin-SHa and NST2, respectively) showing that Gly4 play a moderate role in both activities. When evaluating the effects of two substitutions of Gly to D-Ala, results showed that NST4 with substitution of Gly4 and Gly7 possesses an antibacterial activity similar to NST2 (substitution at Gly4) or NST3 (substitution at Gly7) (MIC values of 12.5, 3.12 and 6.25–12.5 µM for NST2, NST3 and NST4, respectively) but with an higher toxicity to human gastric cells (around 5-folds increase compared to temporin-SHa) (IC_50_ on N87 cells of 1558 ± 324, 1247 ± 240, 520.1 ± 148 and 306.5 ± 48.4 µM for temporin-SHa, NST2, NST3 and NST4, respectively) showing that the sequential loss of Gly4 and 7 is detrimental for its inocuity. NST6 and NST7, in which Gly7 and 10 or Gly4 and 10 respectively were substituted, maintained their antibacterial activity compared to temporin-SHa (MIC values of 3.12 µM for temporin-SHa, NST6 and NST7) but similarly to NST1 with Gly10 substitution, NST6 and NST7 displayed an important increase in toxicity (around 15-folds increase compared to temporin-SHa) (IC_50_ on N87 cells of 1558 ± 324, 104.4 ± 9.3 and 84.69 ± 9.0 µM for temporin-SHa, NST6 and NST7, respectively) confirming the critical role of Gly10 to prevent the toxicity of the peptides. Interestingly, although NST7 with D-Ala substitutions at positions 4 and 10 retained good antibacterial activity against *H. pylori*, NST7L in which L-Ala were used instead of D-Ala lost almost completely its antibacterial activity (MIC of 3.12, 3.12 and 50 µM for temporin-SHa, NST7 and NST7L) but also its toxic effect (IC_50_ on N87 cells of 1558 ± 324, 84.69 ± 9.0 and 2290 ± 762 µM for temporin-SHa, NST7 and NST7L, respectively), showing that the configuration of the Alanine used to replace the Gly is important too. This result confirms what was previously published in a study with temporin-LK1. This peptide has only one Gly at position 7 which was substituted by D-Ala or other L-amino acids. Only D-Ala analogue showed antimicrobial activity in comparison to the natural peptide or the other analogues [23].

Based on the results of the antibacterial testing done on the reference strain *H. pylori* (ATCC 43504) and on the in vitro cytotoxicity assay, more active and safer peptides (i.e., temporin-SHa, NST1, NST2 and NST3) were further tested. Results showed that these peptides were also active against clinical strains of *H. pylori* resistant to one or more conventional antibiotic classically used to treat *H. pylori* infection with MIC values around 6.25–12.5 µM for the best peptides (i.e., temporin-SHa and NST1). Although the MIC values on clinical strains of these peptides were found slightly higher than the ones obtained with the ATCC strain (i.e., 1.5 to 3.25 µM), we were able to demonstrate that this increase in MIC was due to the fetal bovine serum that had to be added to the media to grow the clinical strains and not to an intrinsic lower sensitivity of these strains compared to the ATCC one. The impact of factors physiologically present in the stomach (i.e., acidic pH, pepsin and mucins) on the antibacterial activity of the peptides was then evaluated. It is hard to treat *H. pylori* using AMPs due to the reason that the physiological conditions in the stomach makes it hard to design a stable peptide that could be delivered orally. For that reason, we designed few peptides where one of them showed pepsin resistance and hence was stable in the stomach harsh conditions. Results showed that although the presence of porcine gastric mucins had a low impact on their activity (2-folds increase in MIC), only NST1 showed partial resistance to the digestion by pepsin in acidic environment. For that reason, only NST1 and temporin-SHa (for comparison) were further tested.

In order to evaluate the possible use of temporin-SHa and NST1 to treat patients already infected by *H. pylori*, the ability of the peptides to eradicate established infection has to be measured using time kill assay in condition mimicking the infection conditions, i.e., at pH 6 and in the presence of mucus. It may seem surprising, but although the pH of the stomach lumen is around 1–2, *H. pylori*, through the activity of its urease, is able to increase the pH locally to pH 6 [31]. Thus, activity of the peptide has to be tested at this pH. Similarly, the presence of mucus in the stomach may trap the peptides and although MIC testing demonstrate that mucins only slightly decrease the antibacterial activity of temporin-SHa and NST1, their impact on the kinetic of killing of *H. pylori* had to be evaluated. Time kill assay showed that temporin-SHa and NST1, in accordance with their ability to quickly insert into lipids of *H. pylori* and to cause pore formation, were able to rapidly kill *H. pylori* with 98% and 36%, 99% and 87%, 99% and 99%, and 95% and 100 % killing respectively after 5, 20, 60 and 180 min of incubation, whereas amoxicillin, a conventional bacteriolytic antibiotic classical used to treat patients infected by *H. pylori* could not start its activity before 60 min of incubation reaching 55% after 60 and 180 min of exposure, eventually reaching 75% killing after a 540 min contact. This result confirms what was previously published in a study regarding temporin analogs [21]. Importantly, the presence of porcine (MUC2 and MUC3) or human gastric mucus (N87 cells) only slightly delayed the action of temporin-SHa and NST1 without preventing it. Finally, the use of human tissue obtained from patients allowed the evaluation of the ex vivo toxicity of temporin-SHa and NST1 on human gastric explants. Toxicity was evaluated both in term of cell damages (through LDH assay) and in term of tissue damages (through microscopic observations). Both assays demonstrated that temporin-SHa and NST1, even at 100 µM, had minor to no effect on tissue viability (Figure 9) and tissue architecture (Figure 10 and Figure 11) suggesting their potential safe use to treat patients infected *H. pylori*, including resistant strains.

## 5. Conclusions

Taken together, this study demonstrated that temporin-SHa and its analogs NST1 may be considered as safe alternative to treat *H. pylori* infection, including multi-resistant to conventional antibiotics refractive to actual treatment.

## Figures and Tables

**Figure 1 biomolecules-09-00598-f001:**
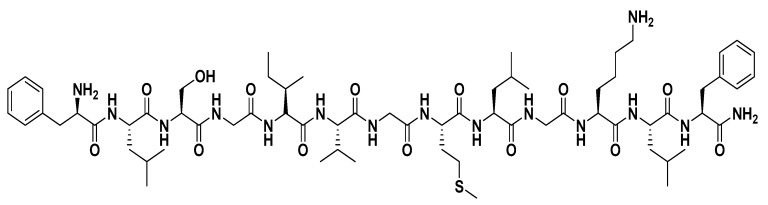
Structure of temporin-SHa.

**Figure 2 biomolecules-09-00598-f002:**
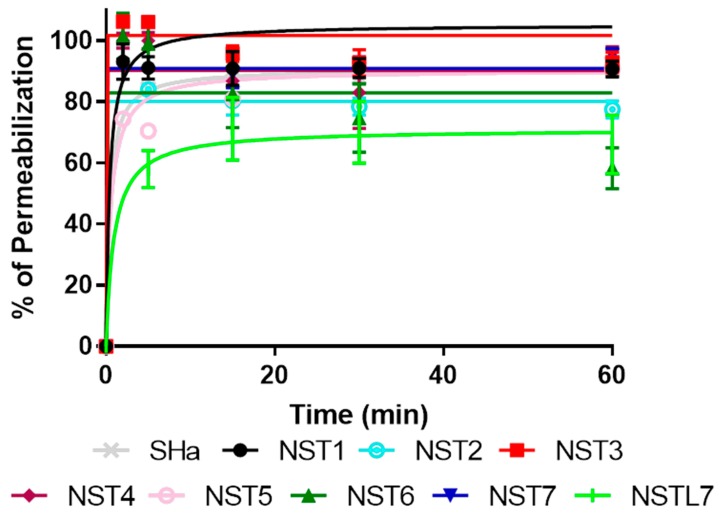
Time-dependent effect of temporin-SHa and its analogs on the membrane integrity of *H. pylori*. *H. pylori* (ATCC 43504) was exposed to temporin-SHa or its analogs at five times their MIC. Kinetics of changes in fluorescent signal (excitation at 530 nm and emission at 590 nm) corresponding to membrane permeabilization and propidium iodide cell entry were measured over 60 min of incubation at 37 °C in micro-aerobic condition. Results are expressed as percentage of membrane permeabilization using CTAB (cetyl-trimethylammonium bromide) at 300 µM as positive control giving 100% permeabilization. Results corresponding to means ± SD (*n* = 3) were fitted using the Graph Pad Prism software.

**Figure 3 biomolecules-09-00598-f003:**
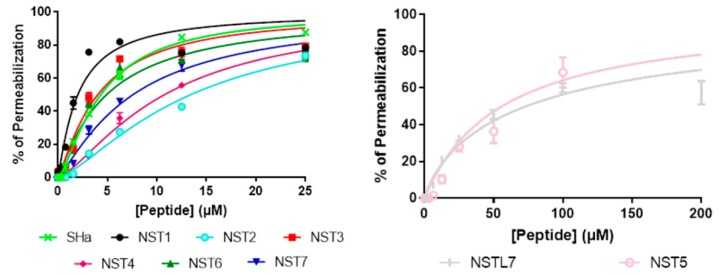
Concentration-dependent effect of temporin-SHa and its analogs on the membrane integrity of *H. pylori*. *H. pylori* (ATCC 43504) was exposed to increasing concentrations of temporin-SHa or its analogs. Fluorescent signal (excitation at 530 nm and emission at 590 nm) corresponding to membrane permeabilization and propidium iodide cell entry was measured after 5 min of incubation at 37 °C in micro-aerobic condition. Results are expressed as percentage of membrane permeabilization using CTAB at 300 µM as positive control giving 100% permeabilization. Results corresponding to means ± SD (*n* = 3) were fitted using the Graph Pad Prism software.

**Figure 4 biomolecules-09-00598-f004:**
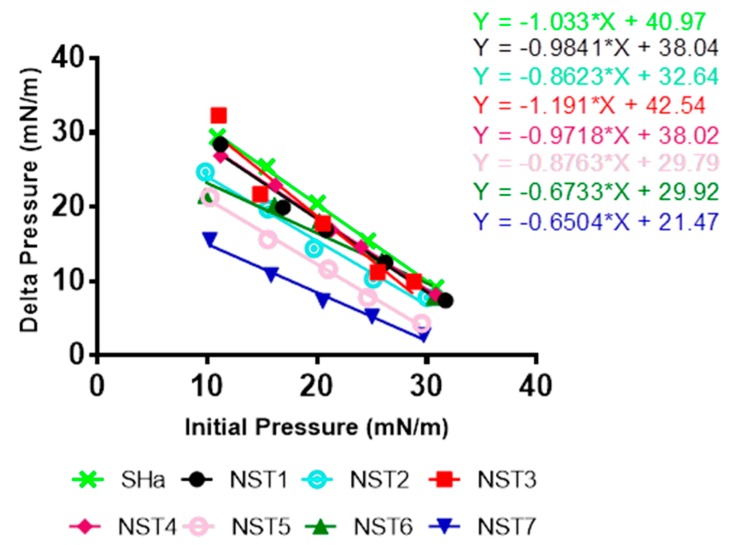
Measurement of the insertion of temporin-SHa and its analogs into total lipids of *H. pylori*. Insertion of temporin-SHa and its analogs into the total lipids of *H. pylori* (ATCC 43504) was studied as explained in Materials and Methods. Data are plotted as Delta Pressure (difference between final and initial pressure) with respect to Initial pressure at injection of peptides.

**Figure 5 biomolecules-09-00598-f005:**
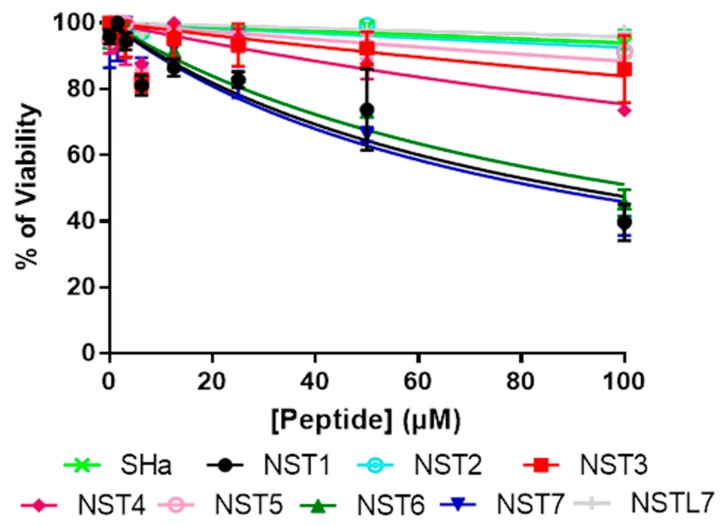
In vitro evaluation of the cytotoxicity of temporin-SHa and its analogs using human gastric cells. Dose-dependent effect of 48 h incubation with increasing doses of temporin-SHa and its analogs on the viability of human gastric cells (N87 cells) was measured using the metabolic activity resazurin assays. Results are expressed as percentage of cell viability with respect to concentration (in µM) of peptides. Curves were fitted using the GraphPad^®^ Prism 7 software. Results were expressed as means +/- SD (*n* = 3).

**Figure 6 biomolecules-09-00598-f006:**
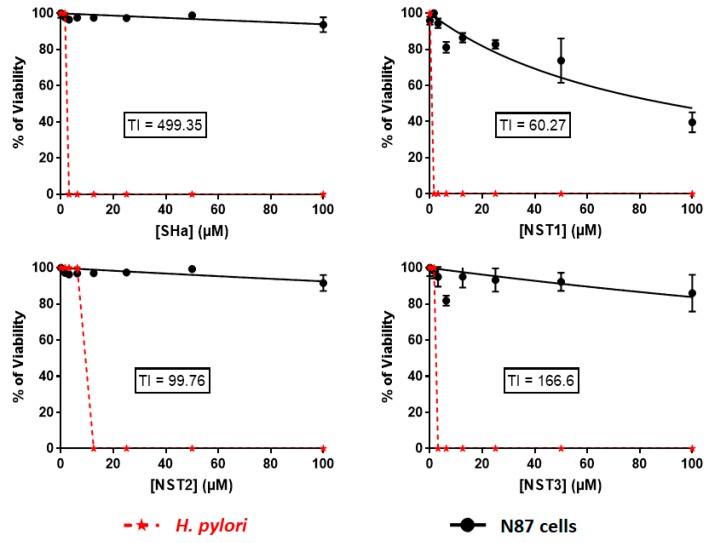
Evaluation of the therapeutic safety of temporin-SHa, NST1, NST2, and NST3. Dose-dependent effect of the peptides on the viability of *H. pylori* (ATCC 43504) and human N87 cells was measured. TI was calculated by dividing IC_50_ values obtained from cytotoxicity curves on N87 by MIC values obtained with *H. pylori* (ATCC 43504).

**Figure 7 biomolecules-09-00598-f007:**
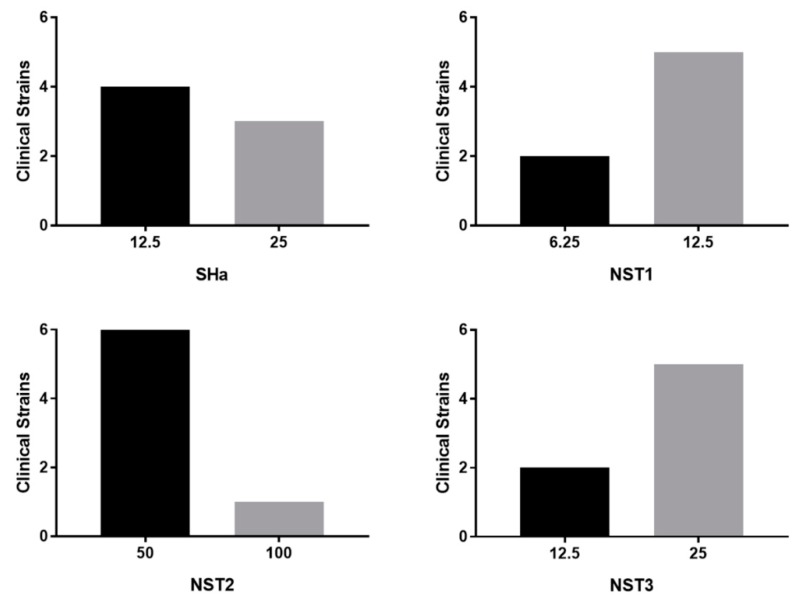
Graphical representation of the repartition of the MIC values of temporin-SHa, NST1, NST2, and NST3 against clinical strains of *H. pylori*. X-axis are MIC values of the peptides in µM. Y-axis indicate the number of clinical strains for each MIC values.

**Figure 8 biomolecules-09-00598-f008:**
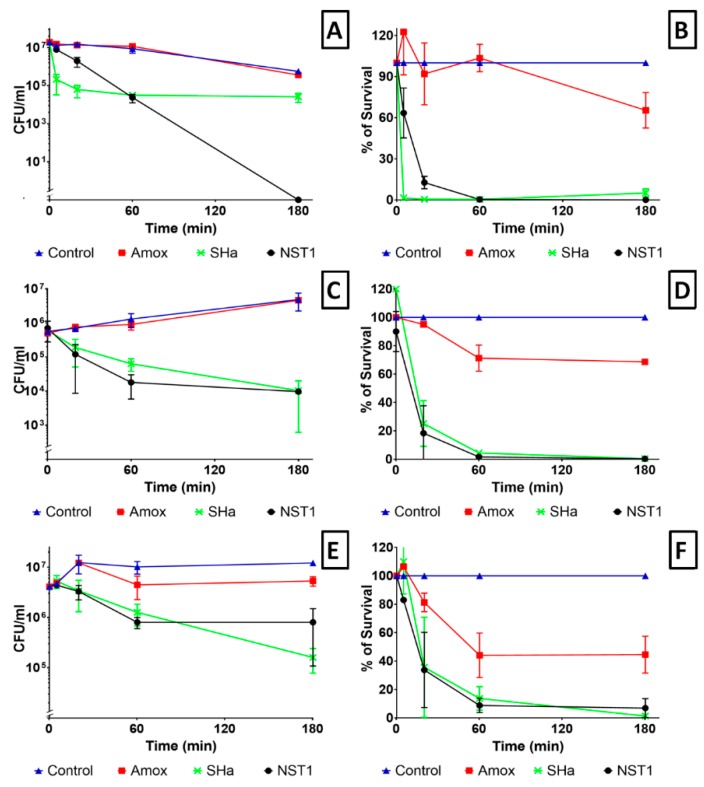
Kinetics of *H. pylori* (ATCC 43504) killing by temporin-SHa, NST1 and the conventional antibiotic amoxicillin. *H. pylori* (ATCC 43504) was exposed for 0, 5, 20, 60 or 180 min to 5-times the MIC of peptides or amoxicillin. Number of surviving bacteria was determined as explained in Materials and Methods. Results are expressed in both in number of colony-forming units (**A**,**C**,**E**) and in % of bacterial survival (**B**,**D**,**F**). Conditions tested were PBS pH 6 (**A**,**B**), PBS pH 6 with porcine gastric mucins MUC2 and MUC3 (**C**,**D**) and PBS pH 6 in the presence of human mucins produced by N87 cells (**E**,**F**).

**Figure 9 biomolecules-09-00598-f009:**
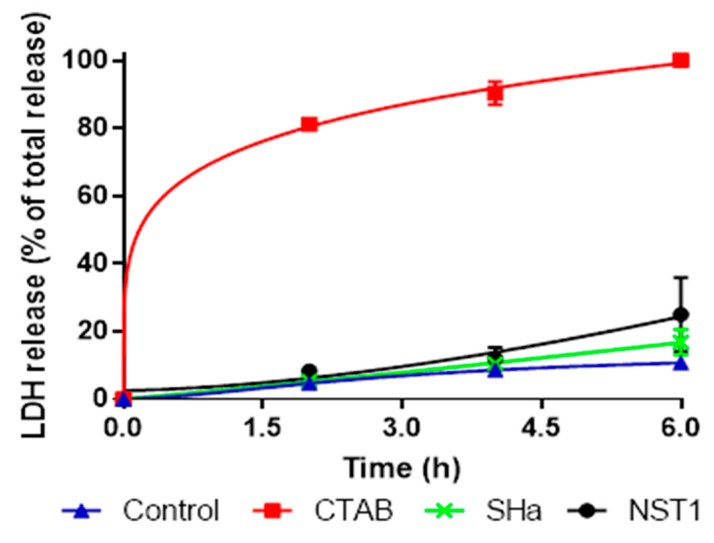
**Lactate dehydrogenase** (LDH) release from human gastric tissue exposed to temporin-SHa or NST1. Human gastric explants were treated with temporin-SHa (100 µM) or NST1 (100 µM) for 2, 4, 8 h before quantification of the release of LDH as explained in Materials and Methods. The LDH release caused by peptides was expressed as percentage of total release obtained with CTAB (300 µM) used as positive control of tissue damages giving 100% release at 8 h. Results corresponding to means ± SD (*n* = 3) were fitted using the Graph Pad Prism software.

**Figure 10 biomolecules-09-00598-f010:**
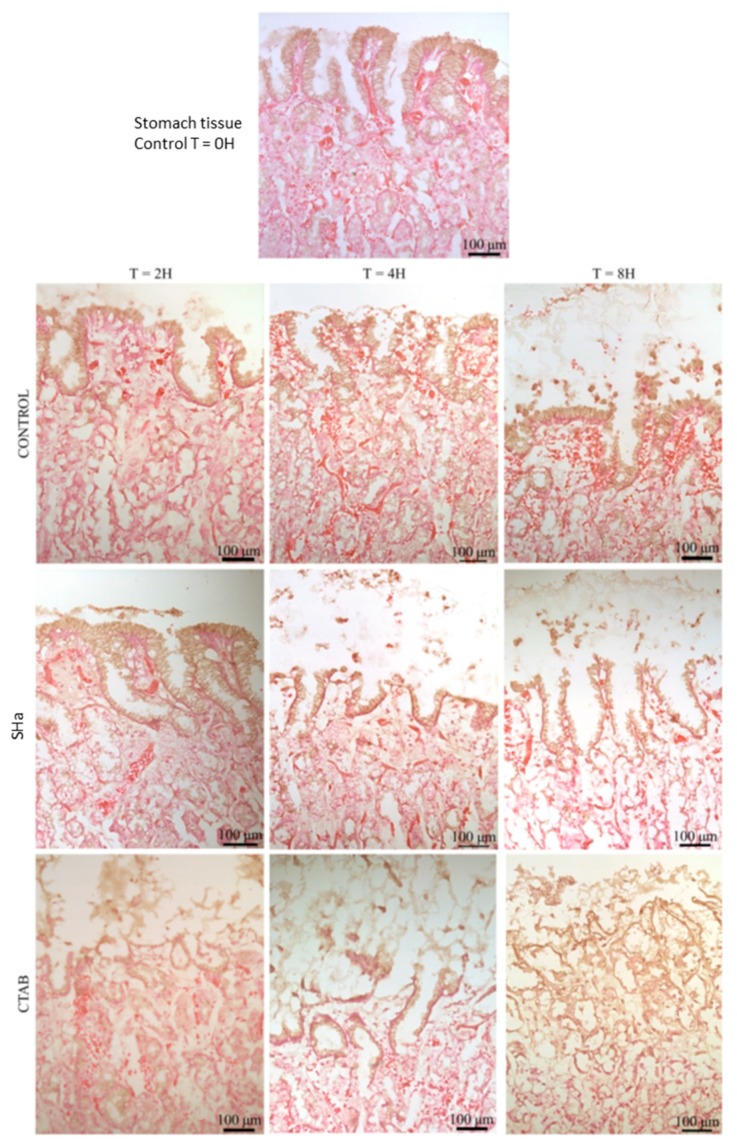
Histological examination of human stomach tissue treated with temporin-SHa. Human gastric explants were treated with temporin-SHa (100 µM) or CTAB (300 µM) as positive control of tissue damages for 2, 4, 8 h before H&E staining and microscopic observations as explained in Materials and Methods. Images shown are representative of effects observed in triplicate.

**Figure 11 biomolecules-09-00598-f011:**
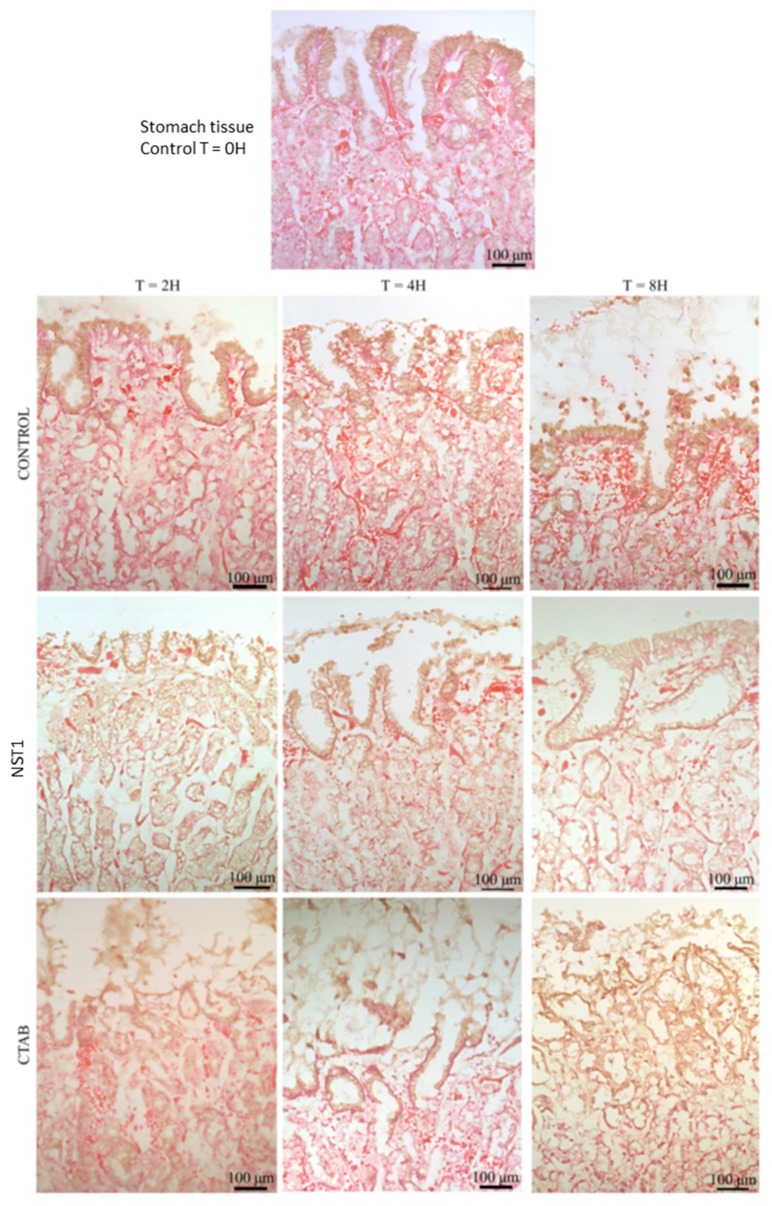
Histological examination of human stomach tissue treated with NST1. Human gastric explants were treated with NST1 (100 µM) or CTAB (300 µM) as positive control of tissue damages for 2, 4, 8 h before H&E staining and microscopic observations as explained in Materials and Methods. Images shown are representative of effects observed in triplicate.

**Table 1 biomolecules-09-00598-t001:** Sequence of peptides used in this study. Temporin-SHa (SHa) is the original peptide. Substitution of one Gly or more is shown in red for D-Ala and in blue for L-Ala. NS = no substitution (natural peptide). G4 = Gly at position 4. G7 = Gly at position 7. G10 = Gly at position 10. a = D-Ala. A = L-Ala.

Peptide	Substitutions	Sequence
SHa	NS	H-Phe^1^-Leu^2^-Ser^3^-Gly^4^-Ile^5^-Val^6^-Gly^7^-Met^8^-Leu^9^-Gly^10^-Lys^11^-Leu^12^-Phe^13^-NH_2_
NST1	[G10a] SHa	H-Phe^1^-Leu^2^-Ser^3^-Gly^4^-Ile^5^-Val^6^-Gly^7^-Met^8^-Leu^9^- D-Ala^10^-Lys^11^-Leu^12^-Phe^13^-NH_2_
NST2	[G4a] SHa	H-Phe^1^-Leu^2^-Ser^3^- D-Ala^4^-Ile^5^-Val^6^-Gly^7^-Met^8^-Leu^9^-Gly^10^-Lys^11^-Leu^12^-Phe^13^-NH_2_
NST3	[G7a] SHa	H-Phe^1^-Leu^2^-Ser^3^-Gly^4^-Ile^5^-Val^6^- D-Ala^7^-Met^8^-Leu^9^-Gly^10^-Lys^11^-Leu^12^-Phe^13^-NH_2_
NST4	[G4,7a] SHa	H-Phe^1^-Leu^2^-Ser^3^- D-Ala^4^-Ile^5^-Val^6^- D-Ala^7^-Met^8^-Leu^9^-Gly^10^-Lys^11^-Leu^12^-Phe^13^-NH_2_
NST5	[G4,7,10a] SHa	H-Phe^1^-Leu^2^-Ser^3^- D-Ala^4^-Ile^5^-Val^6^- D-Ala^7^-Met^8^-Leu^9^- D-Ala^10^-Lys^11^-Leu^12^-Phe^13^-NH_2_
NST6	[G7,10a] SHa	H-Phe^1^-Leu^2^-Ser^3^-Gly^4^-Ile^5^-Val^6^- D-Ala^7^-Met^8^-Leu^9^-D-Ala^10^-Lys^11^-Leu^12^-Phe^13^-NH_2_
NST7	[G4,10a] SHa	H-Phe^1^-Leu^2^-Ser^3^- D-Ala^4^-Ile^5^-Val^6^- Gly^7^-Met^8^-Leu^9^- D-Ala^10^-Lys^11^-Leu^12^-Phe^13^-NH_2_
NSTL7	[G4,10A] SHa	H-Phe^1^-Leu^2^-Ser^3^- L-Ala^4^-Ile^5^-Val^6^- Gly^7^-Met^8^-Leu^9^- L-Ala^10^-Lys^11^-Leu^12^-Phe^13^-NH_2_

**Table 2 biomolecules-09-00598-t002:** Peptide Specifications. tR (retention time); MS (Mass spectrometric); MW (Molecular Weight).

Temporin	Code	tR (min)	MS Data	Chemical Formula	Calculated MW
SHa	-	4.638	HR-MALDI *m*/*z* 1402.7896 [M + Na]^+^	C_67_H_109_N_15_NaO_14_S	1402.7891
[G10a]SHa	NST1	3.489	HR-MALDI *m*/*z* 1394.822 [M+H]^+^	C_68_H_112_N_15_O_14_S	1394.823
[G4a]SHa	NST2	3.673	HR-MALDI *m*/*z* 1416.8048 [M + Na]^+^	C_68_H_111_N_15_NaO_14_S	1416.8053
[G7a]SHa	NST3	3.381	HR-MALDI *m*/*z* 1416.8084 [M + Na]^+^	C_68_H_111_N_15_NaO_14_S	1416.8053
[G4,7a]SHa	NST4	3.467	HR-MALDI *m*/*z* 1430.8204 [M + Na]^+^	C_69_H_113_N_15_NaO_14_S	1430.8204
[G4,7,10a]SHa	NST5	3.501	HR-MALDI *m*/*z* 1444.8361 [M + Na]^+^	C_70_H_115_N_15_NaO_14_S	1444.8366
[G7,10a]SHa	NST6	3.361	HR-MALDI *m*/*z* 1430.8209 [M + Na]^+^	C_69_H_113_N_15_NaO_14_S	1430.8204
[G4,10a]SHa	NST7	3.052	MALDI *m*/*z* 1430.8 [M + Na]^+^	C_69_H_113_N_15_NaO_14_S	1430.8204
[G4,10A]SHa	NSTL7	4.173	MALDI *m*/*z* 1430.8 [M + Na]^+^	C_69_H_113_N_15_NaO_14_S	1430.8204

**Table 3 biomolecules-09-00598-t003:** Evaluation of the antibacterial activity of temporin-SHa and its analogs against *H. pylori* (ATCC 43504) and of their in vitro toxicity against human N87 cells by the inhibitory concentrations reducing 50% of the cell viability (IC_50_ values). Therapeutic index (TI) was calculated by dividing the IC_50_ by MIC.

Peptide	MIC on *H. pylori* (ATCC 43504) (µM)	IC_50_ on N87 Cells (µM)	TI (IC_50_/MIC)
SHa	3.12	1558 ± 324	499.35
NST1	1.5	90.41 ± 10.4	60.27
NST2	12.5	1247 ± 240	99.76
NST3	3.12	520.1 ± 148	166.6
NST4	6.25–12.5	306.5 ± 48.4	24.5
NST5	100	762.1 ± 235	7.62
NST6	3.12	104.4 ± 9.3	33.46
NST7	3.12	84.69 ± 9.0	27.14
NSTL7	50	2290 ± 762	45.8

**Table 4 biomolecules-09-00598-t004:** Evaluation of insertion of temporin-SHa and its analogs into total lipids extracted from *H. pylori*. Total lipids from *H. pylori* (ATCC 43504) were used to study the interaction of temporin-SHa and its analogs with bacterial lipids as explained in Materials and Methods. Values of critical pressure of insertion (CPI) were calculated from the slopes equations in Figure 4 and are expressed in mN/m.

Peptide	SHa	NST1	NST2	NST3	NST4	NST5	NST6	NST7
CPI	39.66	38.65	37.86	35.73	39.12	33.99	44.43	33

**Table 5 biomolecules-09-00598-t005:** Antibacterial activity of temporin SHa, NST1, NST2, and NST3 against resistant clinical strains of *H. pylori*. Clinical strains were found resistant either to clarithromycin (Clari) or metronidazole (Metro) or both. MIC are expressed in µM.

Strain	Resistance	SHa	NST1	NST2	NST3
BM18	Clari/Metro	25	12.5	50	25
LN18	Clari	12.5	6.25	50	12.5
MP78	Clari/Metro	25	12.5	50	25
NM44	Clari/Metro	12.5	6.25	50	25
EM68	Clari/Metro	12.5	12.5	100	25
MF76	Clari/Metro	12.5	12.5	50	12.5
MS67	Clari/Metro	25	12.5	50	25

**Table 6 biomolecules-09-00598-t006:** Evaluation of the impact of mucins or pepsin on the activity of temporin-SHa, NST1, NST2, NST3. The impact of gastric factors (mucus, acidic pH, and pepsin) on the activity of temporin-SHa and its analogs on *H. pylori* (ATCC 43504) was evaluated as explained in Materials and Methods.

Peptide	MIC in the Presence of mucins (µM)	MIC after Digestion by Pepsin (µM)
SHa	6.25	>50
NST1	3.12	25
NST2	25	>50
NST3	6.25	>50

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
