# Peer review of "Temporin-SHa and Its Analogs as Potential Candidates for the Treatment of Helicobacter pylori"

_biomolecules, 2019, doi:10.3390/biom9100598_

Round 1

Reviewer 1 Report

Excellent work.  Congratulations to authors.  during these days it is rare to see paper so complete such this one.   The paper merits publication in Biomolecules.

Few modifications:

Gly and Ala instead of Glycine and Alanine, use three letter code through the text.  In some places, leu, ala, it should be Leu, Ala (the  first letter always capital)

The Figure 1 is very poor and even difficult to read

Table 1, if the N-terminal Phe1 is free indicated in all peptides: H-Phe1-

OxymaPure instead if oxymapure

I suggestion for future work.  Have the authors considered to prepare analogues with more aggresive modifications?  All chenges performed are rather conservative.

Author Response

Comment 1: Excellent work. Congratulations to authors. During these days, it is rare to see a paper so complete such this one. The paper merits publication in Biomolecules.

Answer to comment 1: We would like to thank reviewer 1 for her/his very nice comment about our work.

Comment 2: Gly and Ala instead of Glycine and Alanine, use three letter code through the text.  In some places, leu, ala, it should be Leu, Ala (the first letter always capital)

Answer to comment 2: Accordingly, to reviewer’s suggestion, we changed all amino acid names to 3 letter codes and capitalized the lower cased in the revised manuscript.

Comment 3: Figure 1 is very poor and even difficult to read

Answer to comment 3: Accordingly, to reviewer’s suggestion, we changed the figure to a clearer one.

Comment 4: In Table 1, if the N-terminal Phe1 is free indicated in all peptides, correct to H-Phe1-. Also correct to OxymaPure instead if oxymapure.

Answer to comment 4: Accordingly, to reviewer’s suggestion, we added an H preceding all sequences to indicate free terminal, and we changed oxymapure to OxymaPure as suggested.

Comment 5: I suggest, for future work, that the authors consider preparing analogues with more aggressive modifications?  All changes performed are rather conservative.

Answer to comment 5: Thank you for this important suggestion. We are currently designing new analogues of Temporin with some major changes. The substitutions improved the selectivity of the peptides.

Reviewer 2 Report

The manuscript describes the design and evaluation of biological properties of novel derivatives of Temporin-SHa active against Helicobacter pylori.As a result of the studies, the most promising analog NST1 with the best profile of antimicrobial activity, safety towards eukaryotic cells and resistance to pepsin digestion was selected and proposed as potential anti-H.pyloridrug-candidate. In my opinion, the presented findings will be of wide interest for readers, as H. pylori, in spite of its statues as class 1 carcinogen, is still not the first-choice target in the design of AMP-based antimicrobials, so, not many  articles could be found referring to this subject. The most probable reason of this phenomenon is both poor stability of peptides under conditions of oral delivery and complexity of methodology that should be applied to evaluate H. pylori antibiotic susceptibility in an adequate way. It should be mentioned, that in case of the latter, the authors had coped with the task in impressive way. Presented data was obtained paying a necessary attention to physiological conditions of H. pylori infection and growth requirements of the pathogen. It also describes comprehensive in vitro and ex vivo experimental approaches that could be interesting for all who work on the design of potential novel therapeutics against this bacteria. However, there are a few points that need to be addressed improve the paper for publication.

It is not clear for me, what was the initial goal of substitution of Gly at selected positions 4,7 and 10 with D-Ala or L-Ala.  What kind of structural-activity effect was supposed to be achieved? Maybe it was not properly emphasized in the manuscript and, thus, was unnoticed. The authors mentioned in the paper, that this modification previously appeared successful in induction of anticancer properties of the analogs of temporin-SHa. Was it expected to have the same impact on membranolytic activity towards H. pylori? The reason of this question is that the typical approach in the activity/safety improvement of AMPs usually refers to modifications of structural and physicochemical parameters that lead to the changes in hydrophobicity, amphipathicity, helicity, net charge or, for instance, had a positive impact on serum stability. However, in the manuscript none of these parameters were mentioned as crucial in the applied modification strategy.  Unfortunately, the structural insight into the experimentally observed differences in biological properties between the analogs is also missing in the discussion section.

In my opinion, circular dichroism analysis of a-helical content for compounds in water/TFE mixtures or SDS micelles (or DMPC/DMPG vesicles) could give the answer on the observed difference in their antimicrobial activity/cytotoxicity  or, at least, could give a chance for speculation if helicity is important for these properties or not. It is well known, for example, that D-amino acid substitution in the middle of the amino acid sequence could disrupt alpha-helical structure of the peptide and, thus, substantially affect its antimicrobial activity. It could be interesting for the reader whether it has any effect in case of the proposed temporin-SHa derivatives or not. On the other hand, serum stability or pepsin stability studies supported with LC-MS analysis (instead of MIC evaluation, like in the paper) could give the information about positions within the peptides that are sensitive to degradation, and could be potentially improved. It could also give the information whether the authors deal with enzymatic degradation of the peptides by pepsin or their precipitation in the acidic pH 2.

To sum up, it would be useful for the authors in their future work on AMPs to include this type of technique in their studies. It would be also useful for the quality of the paper if the authors include some information about the rationale of the performed structural modifications.

Page 4: 2.3. Peptide specifications.

I would recommend to perform this data in a form of a table. It would be more convenient for the reader to see the differences between the sequences of the derivatives and their physicochemical properties.

Author Response

Comment 1: The manuscript describes the design and evaluation of biological properties of novel derivatives of Temporin-SHa active against Helicobacter pylori. As a result of the studies, the most promising analog NST1 with the best profile of antimicrobial activity, safety towards eukaryotic cells and resistance to pepsin digestion was selected and proposed as potential anti-H. pylori drug-candidate. In my opinion, the presented findings will be of wide interest for readers, as H. pylori, in spite of its statues as class 1 carcinogen, is still not the first-choice target in the design of AMP-based antimicrobials, so, not many  articles could be found referring to this subject. The most probable reason of this phenomenon is both poor stability of peptides under conditions of oral delivery and complexity of methodology that should be applied to evaluate H. pylori antibiotic susceptibility in an adequate way. It should be mentioned, that in case of the latter, the authors had coped with the task in impressive way. Presented data was obtained paying a necessary attention to physiological conditions of H. pylori infection and growth requirements of the pathogen. It also describes comprehensive in vitro and ex vivo experimental approaches that could be interesting for all who work on the design of potential novel therapeutics against this bacterium. However, there are a few points that need to be addressed improve the paper for publication.

Answer to comment 1: Thank you for comment. We, as suggested, emphasized on the idea that it is hard to treat H. pylori due to the reason that the physiological conditions in the stomach makes it hard to design a stable peptide that could be delivered orally. For that reason, we designed few peptides where one of them showed pepsin resistance and hence was stable in the stomach harsh conditions.

Comment 2: It is not clear for me, what was the initial goal of substitution of Gly at selected positions 4,7 and 10 with D-Ala or L-Ala.  What kind of structural-activity effect was supposed to be achieved? Maybe it was not properly emphasized in the manuscript and, thus, was unnoticed. The authors mentioned in the paper, that this modification previously appeared successful in induction of anticancer properties of the analogs of temporin-SHa. Was it expected to have the same impact on membranolytic activity towards H. pylori? The reason of this question is that the typical approach in the activity/safety improvement of AMPs usually refers to modifications of structural and physicochemical parameters that lead to the changes in hydrophobicity, amphipathicity, helicity, net charge or, for instance, had a positive impact on serum stability. However, in the manuscript none of these parameters were mentioned as crucial in the applied modification strategy.  Unfortunately, the structural insight into the experimentally observed differences in biological properties between the analogs is also missing in the discussion section.

Answer to comment 2: We are pleased by your comment about the rational of our drug design. We are sorry if we were not clear with this issue. We mentioned in the discussion an article (Shah et al. reference 35 … now 26) that studied the effect of substitution of one L-Glycine by D-Alanine for one of the peptides. Temporin-LK1 (the original peptide of the article) was not active whereas the analogue with the D-Alanine showed dramatic increase of activity. For this reason, we showed interest in studying of our peptides against H. pylori. This was added in the introduction

Comment 3: In my opinion, circular dichroism analysis of a-helical content for compounds in water/TFE mixtures or SDS micelles (or DMPC/DMPG vesicles) could give the answer on the observed difference in their antimicrobial activity/cytotoxicity  or, at least, could give a chance for speculation if helicity is important for these properties or not. It is well known, for example, that D-amino acid substitution in the middle of the amino acid sequence could disrupt alpha-helical structure of the peptide and, thus, substantially affect its antimicrobial activity. It could be interesting for the reader whether it has any effect in case of the proposed temporin-SHa derivatives or not. On the other hand, serum stability or pepsin stability studies supported with LC-MS analysis (instead of MIC evaluation, like in the paper) could give the information about positions within the peptides that are sensitive to degradation, and could be potentially improved. It could also give the information whether the authors deal with enzymatic degradation of the peptides by pepsin or their precipitation in the acidic pH 2.

To sum up, it would be useful for the authors in their future work on AMPs to include this type of technique in their studies. It would be also useful for the quality of the paper if the authors include some information about the rationale of the performed structural modifications.

Answer to comment 3: We would like to thank the reviewer for his important suggestion. In future work we will perform such assays to improve the quality of the article as he said.

Comment 4: Page 4: 2.3. Peptide specifications.

I would recommend performing this data in a form of a table. It would be more convenient for the reader to see the differences between the sequences of the derivatives and their physicochemical properties.

Answer to comment 4: Accordingly, to reviewer’s comment, we improved the format of the Peptide specifications section by adding a heading before each paragraph. Also we summarized the data in a small table as recommended.